# Diversity of Arbuscular Mycorrhizal Fungi of the Rhizosphere of *Lycium barbarum* L. from Four Main Producing Areas in Northwest China and Their Effect on Plant Growth

**DOI:** 10.3390/jof10040286

**Published:** 2024-04-12

**Authors:** Yuyao Cheng, Kaili Chen, Dalun He, Yaling He, Yonghui Lei, Yanfei Sun

**Affiliations:** 1Xinjiang Production and Construction Corps Key Laboratory of Oasis Town and Mountain-Basin System Ecology, College of Life Sciences, Shihezi University, Shihezi 832000, China; cyy8823@outlook.com (Y.C.); chenkaili1996@163.com (K.C.); 15806171237@163.com (D.H.); 2College of Medicine, Shihezi University, Shihezi 832000, China; hylxsd@shzu.edu.cn; 3Department of Plant Protection, College of Agriculture, Shihezi University, Shihezi 832000, China

**Keywords:** growth promotion, fungi, diversity identification, enabling benefits

## Abstract

Arbuscular mycorrhizal fungi (AMF) can help plants absorb more mineral nutrients after they colonize plant roots, and the mycelia harmonize the soil structure and physical and chemical properties by secreting compounds. AMF species co-evolve with their habitat’s geographic conditions and hosts; this gradually causes differences in the AMF species. By using Melzer’s reagent to analyze the morphology and using Illumina Miseq sequencing technology to perform the molecular identification of AMF communities among the four typical *L. barbarum* planting areas (Zhongning, Guyuan, Jinghe, and Dulan) investigated, the variety of *L. barbarum* roots and rhizosphere AMF communities was greater in the Zhongning area, and every region additionally had endemic species. The successfully amplified AMF was re-applied to the *L. barbarum* seedlings. We found that the total dry weight and accumulation of potassium increased significantly (*p* < 0.05), and the root volume and number of root branches were significantly higher in the plants that were inoculated with *Paraglomus* VTX00375 in the pot experiment, indicating that AMF improves root development and promotes plant growth. We have investigated AMF germplasm species in four regions, and we are committed to the development of native AMF resources. The multiplication and application of AMF will be conducive to realizing the potential role of biology in the maintenance of agroecology.

## 1. Introduction

*Lycium barbarum* L. (goji berry) is a perennial deciduous plant in the Solanaceae family, and this fruit contains pharmacological components that can provide health benefits, making it an important source of medicine and nutritional supplements [1,2,3]. *Lycium barbarum* has been widely cultivated in China, particularly in the northwest regions [4]. Currently, large quantities of authentically cultivated *Lycium barbarum* exist in such provinces and regions as Ningxia, Gansu, Qinghai, Xinjiang, and Inner Mongolia, with scarce rainfall and infertile land [5]. *Lycium barbarum* is a medicinal herb produced in specific natural conditions and ecological regions; it can adapt to these harsher climatic conditions, and the phenotypic variation in this native herb reflects the result of adaptation to the environment.

The current studies on native *L. barbarum* primarily focus on the effects of environmental factors, such as climatic conditions and soil and water quality, but there is a lack of research on the Arbuscular mycorrhizal fungi (AMF) species of rhizosphere microorganisms and the molecular mechanism of AMF colonization. In addition, long-term cultivation and over-fertilization have led to soil degradation, increases in nitrogen, phosphorus, and potassium, and reduced alkaline phosphatase activity [6], and the phenomenon of farmers digging up old trees and replacing them with young trees of new genetically improved varieties due to their being fewer fruit tree varieties and frequent diseases. The green management model of how to protect the land, while ensuring the economic benefits of *L. barbarum* has become a new and pressing issue.

AMF are key to improving crop productivity and soil quality [7]. After colonizing plant roots, AMF helps plants to expand their root system, and mycelium can also release phosphatase and organic acids [8,9,10] into the soil to dissolve soil nutrients, such as nitrogen, phosphorus, and iron, enhancing soil aggregation and nutrient retention, indirectly affecting soil fertility [11,12,13], and the mycelium of AMF also promotes phosphorus uptake and transport [14]. In addition to this, AMF can be used to improve tolerance and help the host to survive adverse climate changes [15,16,17] and other adverse environmental conditions stresses and foster vegetation recovery in degraded habitats [18].

Comprehensive data on the diversity of species identification and function of AMF must be compiled [19]. Morphological identification allows for the rapid and efficient detection of differences in AMF species in different samples [20]. Data based on nested PCR AMF communities can be interpreted semi-quantitatively, and the degree of species dominance depends mainly on primer selection [21], which is commonly used for AML1/AML2, AMV4.5NF/AMDGR, and NS31/AM1 (Figure 1). At present, no consistent DNA barcode region has been identified for AMF identification [19,22]. This is why many AMF species cannot be accurately categorized into species genera. Different approaches can complement each other to reveal AMF species [23]. It is necessary to associate classical taxonomic evaluations with molecular biological techniques [7], and a combination of two approaches will help to accurately identify AMF communities and also facilitate the improvement of AMF taxonomy [24].

Both plants and animals have different geographical habitats, with varied elevation gradients and soil properties [25,26], and their metabolites and soil extracellular enzymes interact within the microenvironment to alter microbial communities, and thus cause differences in AMF diversity in roots and rhizosphere soils [27]. There is also a selective role of the host plant on its root microorganisms, which divergently selects by interacting with microorganisms, reducing interspecific competition and promoting the coexistence of sister plant species, thus promoting natural selection, facilitating species differentiation and determining species distribution [28]. It can also be argued that AMF communities are the result of co-evolution with local habitats, and thus indigenous AMF species and community distributions are more appropriate for local ecosystems. The use of the already present and colonized AMF species for research and exploitation inhibits the exclusion of alien species. We believe that the species left behind by natural selection must have some distinct advantages, which, together with the artificial regulation of distribution, certainly have good biological control and promotion effects. 

The use of suitable AMF might effectively improve the currently unfavorable situation of economic tree species production on land with saline soil [29], which is beneficial to agroecology due to the use of its biological potential [7]. In addition, capitalizing on ecological processes could improve the functioning of agroecosystems to increase the sustainability of crop yields and mitigate food insecurity [30]. As a Taoist medicinal herb that is the result of specific geographic and ecological conditions, it is worth exploring the amount of influence of AMF has on the growth of *Lycium barbarum*. The growth-promoting effects of different AMF may be different. Therefore, it is of great significance to explore the dominant species that can promote the growth of *Lycium barbarum*.

## 2. Materials and Methods

### 2.1. Sampling Locations and Soil Sampling

Four typical *L. barbarum* cultivation areas in northwestern China were selected as study sites to collect rhizosphere soil and its fine roots (Table 1). The soil was sampled at a vertical distance of 0–15 cm from the target plant and a depth of 20–40 cm to collect soil and roots. Several millimeters from the rhizosphere soil of *L. barbarum* were collected to total 2 kg of each soil sample, and tender roots near the rhizosphere soil were collected whenever possible. Three samples were selected from each site, and each sample was mixed into a single sample using a five-point random sampling method. The soil samples were placed in sterile self-sealing polyethylene bags. The soil that clung to the surface was removed from the samples of collected roots, which were then placed on ice to be taken back to the laboratory.

Each sequencing and field AMF root segment observation treatment was performed in three biological replicates, as were AMF spore identification, multiplication of AMF, and tests of its growth-promoting benefits.

### 2.2. Sample Pretreatment

We air-dried rhizosphere soil using a 20 mm split sieve and stored it at 4 °C for the next step of AMF species identification. One part of the fine roots was cleaned and fixed in formalin, glacial acetic acid, ethanol (FAA) to allow for the later staining of the roots to observe the structure of colonization. Prior to the experiments, juvenile fibrous roots of *L. barbarum* from four regions were collected and stained using trypan blue stain [31] to determine the ability of AMF to colonize the roots of *L. barbarum*. 

Another part of the roots was treatedwith sterile plant tissue surfaces, and the samples were gently rinsed with sterile water for 30 s, cut into sizes of approximately 1 cm, incubated in 70% ethanol for 2 min, placed in 2.5% NaClO contained 0.1% Tween 80 for 5 min, and transferred to 70% sterile ethanol for 30 s. We washed the roots three times with sterile water, and the roots were placed in clean and sterile 1.5 mL centrifuge tubes, flash frozen in liquid nitrogen, and then stored at −80 °C for high-throughput sequencing.

### 2.3. Quantification of AMF Root Colonization

The colonization status of AMF in the root system of *L. barbarum* was observed by staining roots with alkali dissociation-trichothecene blue as previously described. The colonization of roots was observed under an Olympus (CX21; Tokyo, Japan) light microscope, with 10 z as the first level, and 0, 10, 20, etc., in order, with a full level of 100, i.e., the root segment was completely colonized. Zero indicates that is was not colonized. The colonization of various AMF structures was observed and recorded, and the rate of colonization (P*_i_*) was calculated as shown below:$$Pi_{\;}=\frac{\sum i\times N_{i}}{N}\times 100\%$$
where *i* is the colonization grade level of roots; *N* is the total number of roots, and *N_i_* is the number of *i*-level roots.

### 2.4. Extraction of the AMF Spores

The wet sieve method was used to weigh 50 g of the soil sample in distilled water for 20–30 min using a set of clean standard soil sieves with pore sizes of 20 mesh (the aperture of 20 mesh screen is 0.85 mm, which is the same below), 80 mesh (0.18 mm), and 240 mesh (0.06 mm). The sieves were stacked from top to bottom in order of the largest-to-smallest aperture, and the sieve material was collected from the lowest, smallest aperture sieve surface. AMF spores on the sieve surface were rinsed with tap water so that the final volume did not exceed 10 mL, the sieved material was centrifuged using sucrose by placing the sample in a 50 mL centrifuge tube and resuspending it add to 50% sucrose; the final volume did not exceed 50 mL. It was then centrifuged at 3000 rpm for 5 min. The supernatant was poured back onto the sieve with the smallest pore size, and residual sucrose was washed off with distilled water. The sieved material was collected in a 90 cm Petri dish. The modified method of Morton, J. B. was used [32,33].

### 2.5. Morphological Identification of the AMF Spores 

The sieves were sorted by color and morphological type using a Nikon (SMZ800N; Tokyo, Japan) microscope. Mature spores with relatively intact morphology were picked and observed under an Olympus (CX21) light microscope at 400× for characteristics, such as color, shape, size, hyphae, number, type, and the thickness of the layers of spore walls, and color change under Melzer’s reagent. Morphological identification was based on descriptions from websites, such as the International AMF Conservation Center (http://invam.wvu.edu, accessed on 16 December 2021) and the Polish Agricultural University (http://www.zor.zut.edu.pl/Glom-eromycota, accessed on 20 December 2021), with reference to new Chinese species that have been published in recent years. The ecological parameters of the structural diversity of AMF communities are shown in Table 2.

### 2.6. Molecular Identification of AMF

The morphologically distinct AMF spores were screened from the *L. barbarum* rhizosphere soil using wet sieve decantation-sucrosecentrifugation for molecular identification by nested PCR. Individual AMF spores were aspirated under a body microscope, rinsed five times with sterile water, placed in a 1.5 mL centrifuge tube that contained 10 μL Tris-EDTA (TE) buffer with 20% Chelex 100 sodium (Beijing Solarbio Science & Technology Co., Ltd., Beijing, China), and the spores were then mashed thoroughly with a sterile gun, boiled in a water bath for 10 min, incubated on ice for 3–5 min, centrifuged at 10,000 rpm for 2 min, and the supernatant was aspirated into a new centrifuge tube. The extracted DNA was stored at −20 °C.

The target DNA fragments were amplified by SSU rDNA nested PCR [34,35]. The first PCR amplification was performed with extracted AMF monospore DNA in the following reaction system: 2.5 μL 10× PCR buffer, a total of 4 μL dNTPs, 0.1 μL 5 U·μL^−1^ Taq DNA polymerase, and NS1. The reaction procedure involved 94 °C pre-denaturation for 3 min, denaturation at 94 °C for 30 s, annealing at 40 °C for 1 min, extension at 72 °C for 1 min, and extension at 72 °C for 10 min, for a total of 30 cycles. 

For the second PCR amplification, the first PCR product was diluted 100-fold and used as the template. Each dilution was adjusted according to the brightness of the band, and the primers were replaced by AML1 and AML2. The reaction procedure involved 94 °C pre-denaturation for 3 min, 94 °C denaturation for 1 min, 50 °C annealing for 1 min, 72 °C extension for 1 min, and 72 °C extension for 10 min, for a total of 30 cycles. For the third PCR amplification, the second PCR product was diluted 100-fold and used as the template. The primers were replaced by AMV4.5NF and AMDGR. The reaction system was the same as that described above with 0.8 μL each of the AMV4.5NF and AMDGR primers, 1 μL of DNA template, and ddH_2_O supplemented at a total volume of 20 μL. The reaction procedure involved pre-denaturation at 95 °C for 3 min, denaturation at 95 °C for 30 s, annealing at 55 °C for 30 s, extension at 72 °C for 45 s, and extension at 72 °C for 10 min, for 30 cycles. The PCR products were sequenced by Biotech Bioengineering. The resulting sequences were compared with Maarj AM data (https://maarjambo-tany.ut.ee/, accessed on 25 December 2021) data.

### 2.7. Multiplication of AMF Spores

The seeds of *L. barbarum* were sterilized with 2% NaClO_2_ (*v*/*v*) for 10 min. The surfaces were decontaminated with NaClO_2_ (*v*/*v*) for 10 min, sterilized with 75% alcohol for 30 s, and finally rinsed 3–5 times with distilled water. Out of fifty intact AMF spores selected from under the stereoscope (Nikon, SMZ800N), one AMF spore was placed on one *L. barbarum* seed, and peat soil/vermiculite/perlite at 1:1:1 was selected as the culture substrate, which was mixed and repeatedly sterilized twice at 121 °C for 25 min. Gently attached with approximately 1 cm of a wetter substrate, well marked, and placed in an intelligent light incubator (GXZ type; Ningbo Jiangnan Instrument Factory (Ningbo, China)). The light intensity was set to 60% (15,000 Lux), with a light cycle (14 h light/10 h dark) for two days. The initial incubation period was 15 days with sterile water. After all the *L. barbarum* seedlings had sprouted, they were watered once with sterile water during the first week and with a modified version of Hoagland’s nutrient solution (Appendix A) during the second week. The two steps were alternated at 15 mL each time for 6 months after harvesting. Root segments were taken at two-month intervals for infestation surveys.

The roots were studied for trypan blue and alkaline magenta staining to observe the results of colonization and to identify the DNA species of AMF from clones of the infested roots. Clonal identification was conducted using a Plant Genome Extraction Kit (DP305; TianGen Biochemical Technology Company, Beijing, China) and detected by electrophoresis. 

AMF spores from the two successfully propagated samples were subjected to inoculation experiments and were molecularly characterized as *Glo* (*Glomeraceae Glomus Chen14 DL-Glo31*, hereafter G) and *Par* (*Paraglomeraceae Paraglomus* sp. *VTX00375*, hereafter P). The roots of *L. barbarum* seedlings upon which 200–300 spores from each treatment were applied along with colonized root segments were selected, and the a number of AMF spores obtained were selected for inoculation with AMF monocultures and a mixture of the two groups. Three seedlings were left after incubation in each pot, and the four treatments were uninoculated with AMF spores (CK), inoculated with *Glo*, *Par*, and a mixture of *Glo* and *Par*. Each treatment was biologically replicated nine times. Detailed experimental cultivation methods are mentioned earlier in this section; each seedling was watered every 2 days at the beginning of germination depending on soil moisture; water and modified Hoagland nutrient solution was rotated every 5 days after 15 days and weekly after one month until the plants were harvested after 3 months. To ensure that the AMF colonized its roots and did not contaminate the additional strains, three samples were selected from the pots at harvest for clonal identification.

### 2.8. Analysis of Effects by Inoculation on Plant Growth Parameters 

The harvested root samples were gently cleaned and placed in an Expression 1100 XL (Epson; Tokyo, Japan) scanner with forceps, and the roots of each sample plant were scanned to obtain images of each sample. The root analytical system software (WinRHIZO Pro 2013; Regent Instruments, Inc., Quebec City, QC, Canada) was used to measure various parameters of the roots. The number of root branches and the number of root crossings were measured, as well as the number of root tips, root volume, and root length, and the main distribution intervals of these three parameters. The fresh samples were then weighed and placed in an oven for 30 min at 105 °C to kill the roots, which were then dried at 80 °C for 48 h to a constant weight. The samples were finally dried after they had cooled to room temperature. The roots and stems were measured for their dry weight and fresh weight indices. The samples were kept dry and crushed to determine the macroelements.

The crushed and dried samples were first weighed to approximately 0.1 g on an electronic balance and thoroughly heated using fully automatic microwave digestion and extraction apparatus (Mars6; Thermo Fisher Scientific, Waltham, MA, USA), and a plasma emission spectrometer (ICAP6300; Thermo Fisher Scientific, Waltham, MA, USA) was used to determine the Mg (285.2 nm), K (766.4 nm), and Na (589.5 nm) contents.

### 2.9. Statistical Analysis

As the sequencing depth increased, the number of detected sequences leveled off and stopped increasing (Figure 2a). After the taxonomic annotation of the operational taxonomic unit (OTU) of AMF in the root segment, information on the species abundance of the OTU in each sample was obtained and drawn according to the minimum sample sequence. The OTUs that were 97% similar were selected to compare the differential species and shared species in a Venn diagram, and the between-group differences were tested by analysis of similarity (ANOSIM). Principal coordinate analysis (PCA) was performed at the species level using the Bray–Curtis distance algorithm to explore the similarity or difference in AMF community composition among the different sample subgroups. The top 10 species that were the most abundant were selected at the genus level, and the community composition and species abundance distribution in the four regions were counted using heatmap plots. The top 20 species that were the most abundant were selected for analysis of species evolution in the four regions using the maximum parsimony method. Statistics and graphing were performed using R language tools (Rstido, R4.1.2).

The data of AMF colonization rate, growth index, and elemental uptake of *L. barbarum* were analyzed by one-way analysis of variance (ANOVA) using SPSS 20.0 (IBM, Inc., Armonk, NY, USA), and multiple comparisons were performed using Duncan’s method (*p* < 0.05) and plotted with Excel 2019 (2305 Build 16.0.16501.20074).

## 3. Results

### 3.1. Species Diversity of AMF in Roots of L. barbarum

A high overall rate of mycelial colonization of the samples by AMF was identified by observing the root segments with better decolorization (Figure 3). More hyphae, vesicles, and arbuscular structures were produced in Dulan than in the other regions (Table 3), while the rate of vesicle formation in Zhongning was lower and significantly lower than those in the other three regions.

The four areas in which the NingQi1 variety of *L. barbarum* was cultivated were sampled with three replicates per area for a total of 12 samples. The number of OTU species obtained for each sample increased less with an increasing sequencing depth, and when the curve flattened out, it indicated that the sequencing depth was sufficient (Figure 2a). The diversity data analysis of 12 samples from 12 root segments of *L. barbarum* were completed, and 180,130,095 bases of optimized sequences were obtained, with an average sequence length of 215 bp. The species annotation results were counted as one domain, one kingdom, one phylum, one class, five orders, five families, five genera, fourteen species, and sixty-three OTUs. All the phylum levels were Glomeromycota, and the genus level included Glomus_f__Glomeraceae, unclassified_c__Glomeromycetes, Diversispora, Paraglomus, and unclassified_o__ArchaEosporales (Table 4). The OTU sequences that were measured differed significantly among the different samples from the same region when they were combined with the sample dilution curves. For example, one sample from Zhongning contained many more sequences than those from all the other samples. It was apparent that the abundance and diversity of AMF in the root segments showed large differences, and the comparison showed that the values of diversity associated with the root segments in Zhongning were higher than those in the other regions. 

Additional analysis revealed that the numbers of AMF OTUs in the root segments of *L. barbarum* were the highest in Zhongning compared with those in the other four regions, while the other three regions had similar numbers of OTUs and fewer numbers compared with those in Zhongning. At the species level, *unclassified_g__Glomus_f__Glomeraceae* was the species with the highest abundance in the sequenced sequences that were shared by the four regions (Figure 2b), which accounted for Zhongning (66.55%), Jinghe (87.52%), Dulan (89.89%), and Guyuan (50.14%). In addition to this, there were also *unclassified_c__Glomeromycetes* (33.37%), *Glomus-*Wirsel-OTU16-VTX00156 (7.29%), and Glomus-sp.-VTX00304 (4.80%) in the Jinghe area and *Glomus-*MO-G23-VTX00222 (9.05%) and *Glomus-intraradices-*VTX00105 (44.54%) in the Guyuan area.

The community composition of the different samples from the four regions was analyzed by PCA that reflected the degree of variation among the samples (Figure 4a). The samples were mostly clustered in the four regions, except for two, including GY2 (Guyuan) and ZN2 (Zhongning). This condition of large variation in individual samples is reflected in the number of OTUs measured in previous samples, and the two axes explained more than half of the AMF community differences and distances (Bray–Curtis: PC1 = 27.55% and PC2 = 23.90%).

The phylogenetic positions of the top 20 OTUs of the AMF species based on their abundance in the four regions were determined using a phylogenetic tree (Figure 5), and the OTUs of AMF in the root segments of *L. barbarum* were widely distributed in the phylogenetic tree. From the perspective of molecular evolution, the affinities of the species in the samples during evolution were revealed, and the four regions had relatively close overall affinities. Combined with the heatmap, it was apparent that the AMF community species in the roots of *L. barbarum* in Jinghe and Dulan were closely related (Figure 4b). Among them, OTU1 was *Glomus_f__Glomeraceae*, which had a large number of reads in all the four regions. The other genera had fewer OTU sequences and fewer species, and the genus *Diversispora* was unique to the Guyuan region.

### 3.2. Species Diversity of Rhizosphere AMF in L. barbarum

The morphological identification and statistics of AMF in the rhizosphere soil of *L. barbarum* from four regions showed that there were no significant differences in spore density and the number of AMF spores in the four regions (Table 5), which were identified as four orders, six families, eight genera, and forty-three species (Appendix A and Appendix A). The eight genera included *Glomus*, *Rhizophagus*, *Septoglomus*, *Scutellospora*, *Acaulospora*, *Diversispora*, *Paraglomus*, and *Ambispora. Glomus* was found in all the four regions and was the dominant genus in the rhizosphere soils. The genera *Scutellospora*, *Acaulospora*, and *Ambispora* were also common. Four species, including *G. etunicatum*, *G. globiferum*, *Acaulospora bireticulata*, and *Ambispora jimgerdemannii*, were shown to occur more frequently in the sampled sites based on the IF and RA values and varied among the sites sampled. 

By identifying the species based on morphological identification, 50 AMF spores were selected from the Zhongning region, and 25 AMF spores from each of the other three regions were selected for nested PCR molecular identification. A total of 46 valid sequences were obtained, which were classified into eight genera and sixteen species by sequence comparison (Table 6). *Scutellospora*, *Acaulospora*, *Diversispora*, *Paraglomus*, *Ambispora*, *Archaeospora*, and *Glomus* were not identified in the Guyuan and Dulan regions, while *Paraglomus* was found in all the four regions with a frequency of occurrence of more than 10%.

### 3.3. Post-Expansion Result Test

A total of 50 samples were observed with both the stains, of which the number of AMF-colonized samples accounted for 32.0% of the total number of samples, and the AMF was observed mainly in the vesicular structures of the roots of the samples (Figure 6a–c). Two of the samples had a high rate of AMF colonization and excellent infestation, with a large number of AMF spores in the culture medium.

The AMF spores in two sample matrices were separated by the sucrose wet sieve method and found to contain approximately 200–300 AMF spores/10 g of substrate, which were molecularly identified as *Glo* (Glomeraceae *Glomus* Chen14 DL-Glo31) and *Par* (Paraglomeraceae *Paraglomus* sp. VTX00375) bulk AMF spores. The roots of the *L. barbarum* seedlings were reinoculated with single and mixed bulk AMF spores, and the roots were stained at harvest and clonally identified as *Glo*, *Par*, and *Glo* × *Par* (mixed AMF spores). The results were consistent with the initial inoculation of AMF species.

### 3.4. Effect of Inoculation with AMF on the Biomass, Root, and Aboveground mass Element Uptake of L. barbarum

The number of root tips of *L. barbarum* seedling samples under the *Par* treatment was particularly significantly higher than the other treatments in the range 0 < T ≤ 0.5 cm, and the number of root branches and root crosses showed the same trend (Figure 7a). The root volume in the range of 0 < V ≤ 0.5 cm^3^ and the root length in the range of 0.5 < L ≤ 1.0 cm were significantly higher than those of the other samples (Figure 7b,c), which indicated that the seedlings had well-developed root systems, and the roots grew better under this treatment. It is notable that the root volume and root branching under the *Glo* and *Glo* × *Par* treatments were not different from the CK control in terms of crossover, but were lower than the control in terms of root length. In addition, the monitoring of macroelements revealed that all the samples from the AMF inoculation treatments had increased levels of potassium accumulation in the *L. barbarum* seedling plants (Figure 7d), and the potassium content of *Lycium barbarum* under *Par* treatment was 29.20% higher than that of CK (Appendix A). The dry and fresh root weights as well as the total dry weights of *L. barbarum* under the P treatment were significantly higher than those of the other treatments (Figure 7e,f); compared with CK, the dry weight of *Lycium barbarum* root under *Par* treatment increased by 100%, the fresh weight of root increased by 104.76%, and the total dry weight increased by 42.71% (Appendix A). This suggests that inoculation with AMF helps *L. barbarum* to take up a large amount of potassium, which is required for growth and development, and further increases the total dry weight and that of the seedling roots, and thus the relative content of potassium in the plant. The developed extension of the root system was improved by the different AMF species and the varied inoculation methods.

## 4. Discussion

Ningqi 1, a variety of *L. barbarum*, is grown in four different northwestern production areas, and the best overall index of all the active components of the fruit was found in the Zhongning region (Ningxia Province) [5,36]. But the Zhongning region, with the most alkaline and nutrient-poor soil, has managed to become a major production area for *L. barbarum* [37]. This increased adaptation to environmental stress cannot be separated from the gain of microorganisms in their rhizosphere, particularly AMF, which aid *L. barbarum* to some extent by mineralizing organic matter and expanding the surface area of the root system, making its root nutrients available to the plant. In this study, structures such as vesicles, hyphae, and Arum-type structures were observed in the stained roots of *L. barbarum*, indicating that AMF can colonize *L. barbarum* and establish better symbiosis with it. The AMF colonization rates of the roots in the four regions ranged from 60.83% to 98.13%, with a high level of overall colonization. 

The second-generation high-throughput amplicon sequencing of AMF communities in the root of *L. barbarum* from the four regions resulted in the genus *unclassified_g__Glomus_f__Glomeraceae* being the dominant genus and the one with the highest content of OTUs in the roots of *L. barbarum*. The same species of *L. barbarum* is found in different production areas in northwestern China, but each area contains AMF species that are unique to that area and can also play a role in colonization, which is a major factor in the source of variation between the samples. Different AMF colonize their hosts with corresponding benefits, and those variable AMF species in the four regions could help the host to adapt to local geographic and ecological conditions, i.e., the result of co-selection caused by the environment and the host plant [38,39,40]. 

In this study, we used second-generation Illumina MiSeq technology (Illumina, San Diego, CA, USA) to select AMV4.5NF and AMDGR as primers for the second amplification in the soil AMF community detection to find a large number of unclassified OTUs. Owing to the limited and biased sampling of the AMF taxa, the molecular databases do not represent the breadth of AMF diversity, which limits the utility of database-matching approaches [41]. The ITS2 region of the 5.8S rDNA of the 35S rRNA gene, which contains both conserved and variable regions, has been sequenced and found to be highly efficient for identifying AMF using a molecular genetic approach [42]. The sequencing of multiple segments rather than one region for identification is also efficient; the SSU rDNA, LSU rDNA, and ITS regions are important to sequence. SMRT sequencing can be used to identify soil and root AMF communities [41,43]. In addition to this, rapid, accurate, and inexpensive molecular mass determination and automation render matrix-assisted laser desorption/ionization time-of-flight mass spectrometry (MALDI-TOF-MS) are promising alternatives to identify AMF by morphological and molecular methods [44]. A previous study found that most AMF communities in the rhizosphere soil of *L. barbarum* were not identified and assigned to specific species [37] by high-throughput sequencing. This study identified most of the AMF communities in their roots and assigned them to specific species by high-throughput sequencing. To improve the accuracy of the results, the AMFs in the rhizosphere soil of *L. barbarum* were studied in more detail using morphological and molecular methods to identify them. 

A combination of two types of morphological and molecular identification indicated that the rhizosphere of *L. barbarum*. *Glomus* and *Paraglomus* were found in the roots and rhizosphere soil of *L. barbarum*, and *Glomus* accounted for a higher proportion. Some studies have indicated that the family Glomeraceae to which *Glomus* belongs dominates the AMF species in arid areas [45]. *Glomus* that were identified in this study were often detected in the soils of other regions [46,47,48]. The spores of the genus *Glomus* under AMF species are tiny in size, but numerous, leading to widespread distribution and consequent dominance [49], and these spores are more adaptive in adapting spore formation patterns to different environmental conditions [50]. Species that can adapt to a habitat, proliferate and subsequently become dominant, and conversely those that cannot adapt are slowly eliminated to the point of disappearing from that type of habitat. The AMF species that were found at a higher frequency during the experimental observations and had important values could have been present owing to their own biological characteristics that are better adapted to the habitat [51] as a result of evolution in their geographical environmental conditions and host plant co-selection. 

In a long-term evolutionary process of reciprocal symbiosis, native mycorrhizal fungi often provide more effective assistance for phytoremediation [52]. The suitability of AMF species to the host plant was ensured by the single-spore expansion of native AMF screened at the rhizosphere of *L. barbarum*, and it was then applied to *L. barbarum* to test its effect on promoting growth. This study found that the *Glo*, *Par*, and *Glo* × *Par* inoculation of *L. barbarum* produced different effects. The use of locally available AMF in ecosystem restoration experiments is a potentially effective approach. For example, the use of native AMF communities in deserts can provide a local advantage to the growth of desert plant Korshinsk pea shrub (*Caragana korshinskii*) [53]. Additionally, leek seedlings (*Allium tuberosum*) inoculated with AMF were vigorous, resistant to pathogens and water stress [54], and the average shoot fresh weight (i.e., yield) of the leeks increased by 794% [55]. AMF can also be used to improve the production of crops, particularly their quality. Other studies have shown that the quality of tomato (*Solanum lycopersicon*) fruit as determined by the concentration of soluble solids or color improved when inoculated with AMF, and the dose of fertilizer used was lower than that normally used on farms [56]. This indicates that the use of AMF in agriculture facilitates to goal of green food and environmentally friendly products.

The development of plant roots increased under the *Par* treatment, but this was suppressed when it was co-inoculated with other AMFs. The root system was less developed under the G treatment, which suggested that the different treatments and the combination of the two inoculations also affected inoculation effects. However, the root dry weight and total dry weight of the plants were higher than those of the control group, suggesting that the AMFs still promoted the growth and development of the plants to some extent. It is possible that the increase in root mass was due to an increase in the storage of total nonstructural carbohydrates (TNCs) rather than to an increase in the storage mass or root surface area, and in the theory of optimal allocation for plant survival, the biomass of aboveground part is often sacrificed for the most growth-limiting resources [57]. Biomass allocation is not only influenced by ontogeny, and therefore, tree size [58]. It follows that the type of AMF inoculated, the method of inoculation, and the longevity status of the host may influence the outcome of the study. *L. barbarum* is a perennial shrub; in this experiment, we simulated the soil climatic conditions in Northwest China to reduce the input of nutrients and the number of times of watering, so it also showed a trend of slow growth. The overall development of the root system under the *Par* treatment improved, and later in a field trial, the aim may be to observe that the intervention of the complex and variable external conditions can help the plant to improve these performances.

Chen et al. [37] showed that the AMF mycelium in the rhizosphere of the host *L. barbarum* in the Zhongning area with poor soil conditions can secrete phosphatase to convert organic phosphorus into available effective phosphorus since the AMF mycelium can secrete phosphatase. Additionally, we found that inoculation with AMF increased the content of potassium ions in *L. barbarum* seedlings, and AMF may promote cell membrane stability by increasing the uptake of potassium ions and the production of antioxidants [59,60]; inoculation with AMF improves the resistance of *L. barbarum* to coping with low rainfall and infertile soil conditions in Northwest China. 

AMF can affect the host by changing plant succession and distribution, nutrient uptake, developmental growth, resistance, and the quality and yield of target products [61,62,63]. This study provides a perspective for the next exploration of the differences in the ability of specific AMFs to regulate an environmental indicator, which can be achieved by artificially constructing compounded multiple AMFs in different habitat conditions to help plants improve their survival ability. The use of microbial resources increases the crop yield, reduces the input of chemical fertilizer, and improves the product quality. The harmful effects to human health and the environment from the application of fertilizer strongly suggests the urgent need for an environmentally friendly alternative to meet the food needs of the growing world population [64]. 

## 5. Conclusions

In general, we carried out the morphological identification and molecular identification of AMF in four areas of Zhongning, Guyuan, Jinghe, and Dulan, and determined that the colonization rate of AMF in the four areas was relatively high, amongst which the root and rhizosphere AMF community diversity in *Lycium barbarum* in Zhongning was the highest. *Glomus Chen14 DL-Glo31* and *Paraglomus* VTX00375, which are successfully amplified AMF, were used to deal with *Lycium barbarum* seedlings alone and in combination. The results showed that the total dry weight and potassium accumulation of *Lycium barbarum* treated with *Paraglomus* VTX00375 were significantly increased, and the root development of *Lycium barbarum* was significantly promoted. This study laid the basis for exploring the unique germplasm resources in Northwest China and enhancing the AMF germplasm financial institution for its development and utilization. Increasing the crop yield by reducing fertilizer input and using microbial resources (AMF) is critical to the economic and environmental sustainability of crop production.

## Figures and Tables

**Figure 1 jof-10-00286-f001:**
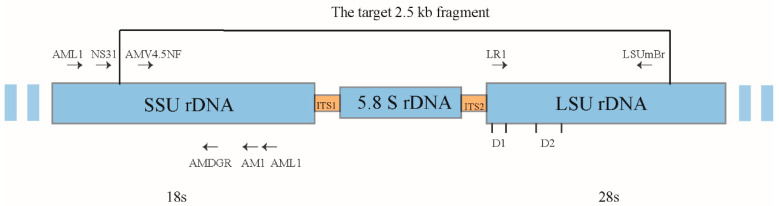
Schematic diagram of the primer region.

**Figure 2 jof-10-00286-f002:**
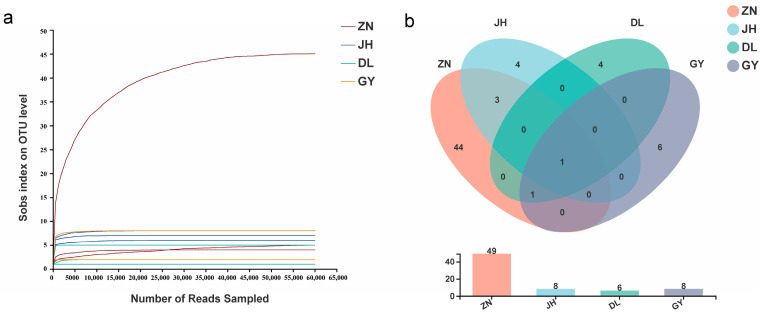
AMF OTU differences in the roots of *L. barbarum* (goji berry) from four regions. (**a**) represents the dilution curves of the root samples from the four regions; (**b**) is the AMF Venn diagram based on operational classification units (OTU). The abbreviations of each county in the figure are Zhongning (ZN), Guyuan (GY), Jinghe (JH), and Dulan (DL), which are the same as shown below. AMF, arbuscular mycorrhizal fungi.

**Figure 3 jof-10-00286-f003:**
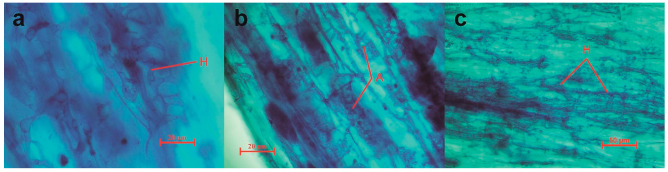
Staining results of different structures were observed for four regions of goji berry (*L. barbarum*) root segments. (**a**–**c**) show different magnifications, respectively. H shows AMF hyphae; A shows a typical AMF infestation of AMF produced after the Arum-type structure. AMF, arbuscular mycorrhizal fungi.

**Figure 4 jof-10-00286-f004:**
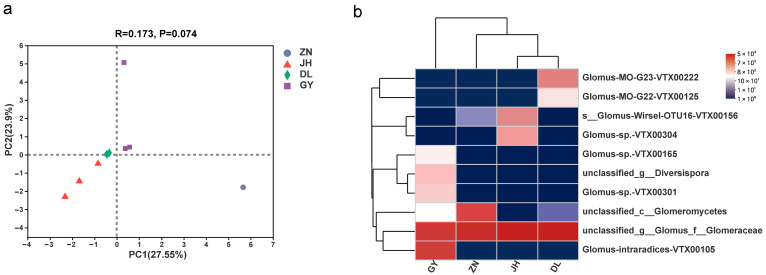
Differences in the diversity of AMF samples of goji berry (*L. barbarum*) roots from four regions. (**a**) shows a principal component analysis (PCA) based on Bray–Curtis distances that shows the differences in AMF communities in different samples; (**b**) shows the statistical species abundance of each sample at the species level in a heatmap. AMF, arbuscular mycorrhizal fungi.

**Figure 5 jof-10-00286-f005:**
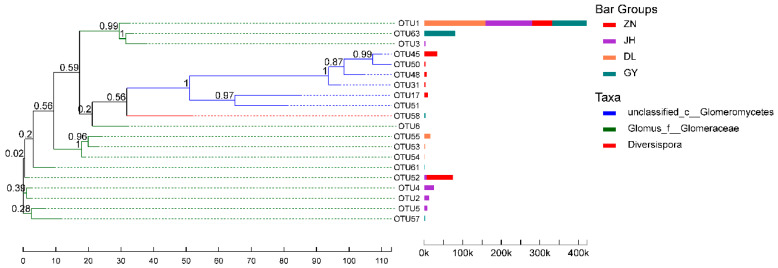
Approximate maximum-likelihood phylogenetic tree. Phylogenetic tree of the OTU levels of AMF from four different habitats. The left side of this figure represents the top 20 species that were the most abundant in terms of proximity, and the right side represents the number of reads corresponding to each species in the four regions. AMF, arbuscular mycorrhizal fungi; DL, Dulan; GY, Guyuan; JH, Jinghe; ZN, Zhongning.

**Figure 6 jof-10-00286-f006:**
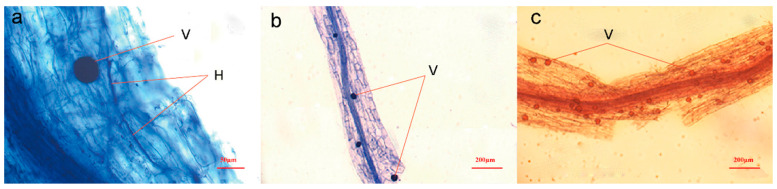
Identification of colonization by *L. barbarum* seedlings. (**a**,**b**) show trypan blue staining, and (**c**) shows alkaline magenta staining. H: hyphae, V: vesicle.

**Figure 7 jof-10-00286-f007:**
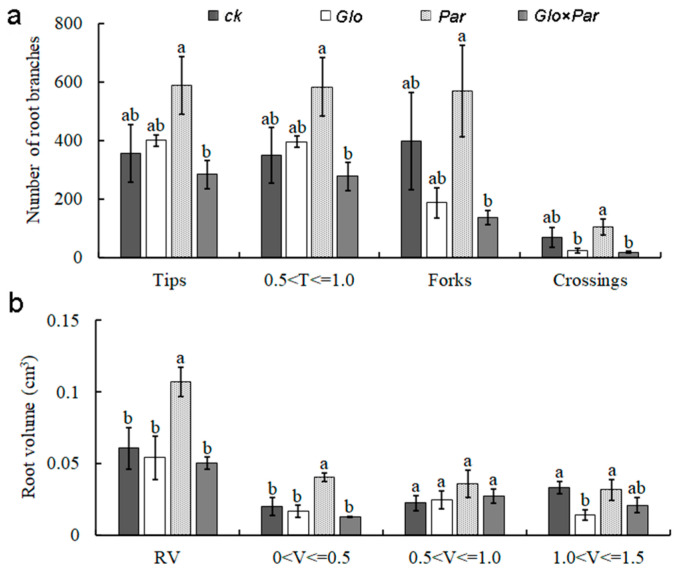
The effect of inoculation with AMF on the biomass of *L. barbarum*. The inoculated *Glomus Chen14 DL-Glo31* is abbreviated as *Glo*, *Paraglomus* sp. *VTX00375* is abbreviated as *Par*, and the mixture of the two is *Glo*×*Par*; CK is the control. *p* < 0.05. (**a**–**c**) represent the determination of the root ecological parameters index. Root tip length interval (cm), 0.5 < T ≤ 1.0. Root volume interval (cm^3^): 0 < V ≤ 0.5, 0.5 < V ≤ 1.0, 1.0 < V ≤ 1.5. Root length interval (cm), 0.5 < L ≤ 1.0. (**d**) represents the uptake of massive elements by the aboveground parts of the seedlings after AMF inoculation. (**e**,**f**) represent dry weight and fresh weight, respectively. RDW, root dry weight, SDW, shoot dry weight, TDW, total dry weight, RFW, root fresh weight, SFW, shoot fresh weight, TFW, total fresh weight.

**Table 1 jof-10-00286-t001:** The geographical location, environmental condition, and soil type of sample collection.

Province	Location (County)	Soil Type	Acquisition Subjects	Sample Codes	Latitude (N)	Longitude (E)	Altitude (m)	Average Annual Precipitation (mm)	Average Annual Temperature	Soil Characteristics
Ningxia	Zhongning	Calcisols soil	*Lycium barbarum* (NingQi 1)	ZN1 ZN2 ZN3	37°16′26″	105°27′16″	1123	202.1	9.5 °C	Oasis soil
Ningxia	Guyuan	Chernozems Soil	*Lycium barbarum* (NingQi 1)	GY1 GY2 GY3	36°25′48″	106°9′	1429	516.7	8.3 °C	Red clay
Xinjiang	Jinghe	Kastanozems soil	*Lycium barbarum* (NingQi 1)	JH1 JH2 JH3	44°20′28″	82°32′7″	290	129.0	8.0 °C	Calcium palm fiber
Qinghai	Dulan	Leptosols soil	*Lycium barbarum* (NingQi 1)	DL1 DL2 DL3	36°13′37″	92°15′19″	2783	179.1	2.7 °C	Calcium palm fiber

Note: The soil types were obtained from https://www.fao.org/soils-portal/data-hub/soil-maps-and-databases/harmonized-world-soil-database-v12/en/, accessed on 9 April 2024. NingQi 1 is a cultivar of *Lycium barbarum*. The abbreviations of each county shown in the figure are Zhongning (ZN), Guyuan (GY), Jinghe (JH), and Dulan (DL).

**Table 2 jof-10-00286-t002:** Ecological parameters of the structural diversity of AMF communities.

Ecological Parameters	Explanation
Spore density (SD)	Refers to the total number of AMF spores contained in each 50 g of air-dried soil sample.
Species richness (SR)	The total number of AMF spore species per 50 g of soil sample
Isolation frequency (IF)	The proportion of a genus or species of AMF that was present in the overall sample.
Relative abundance (RA)	The proportion of a genus or species of AMF in the total number of spores at a sample site.
Importance value (IV)	Refers to the average of separation frequency and relative abundance.
Diversity index: Shannon–Wiener index (H) and Simpson’s diversity index (D)	$H=-\sum P_{i}\ln P_{i}$ and $D=1-\sum \left({Pi}^{2}\right)$, where $P_{i}=n_{i}/N$; $n_{i}$ is the number of AMF spores of a certain species (genus) in a sample site, and N is the total number of AMF spores in this sample site.

Note: The importance value IV was used to classify the AMF dominance into three classes: IV > 30%, dominant species (genus); 10% < IV ≤ 30%, common species (genus); and 0% < IV ≤ 10%, rare species (genus). AMF, arbuscular mycorrhizal fungi.

**Table 3 jof-10-00286-t003:** AMF colonization and rhizosphere soil AMF diversity.

Region	Hyphae	Vesicles	Arbuscules	Total Colonization Rates
ZN	68.57%	2.86%	14.29%	68.57%
GY	60.50%	36.83%	16.17%	60.83%
JH	82.92%	46.00%	18.46%	82.92%
DL	98.13%	37.88%	36.13%	98.13%

Note: The abbreviations of each county shown in the figure are Zhongning (ZN), Guyuan (GY), Jinghe (JH), and Dulan (DL). AMF, arbuscular mycorrhizal fungi.

**Table 4 jof-10-00286-t004:** Identification of AMF OTU species in *Lycium barbarum* roots.

Phylum (1) Glomeromycota Class (1) Glomeromycetes Orders (5)	Families (5)	Genera (5)	Species (14)
unclassified_c_ Glomeromycetes	unclassified_c_ Glomeromycetes	*unclassified_c_* *Glomeromycetes*	*unclassified_c__Glomeromycetes*
Glomerales	Glomeraceae	*Glomus_f_* *Glomeraceae*	*unclassified_g__Glomus_f__* *Glomeraceae*
			*Glomus-Wirsel-OTU16-VTX00156*
			*Glomus-sp.-VTX00304*
			*Glomus-viscosum-VTX00063*
			*Glomus-MO-G22-VTX00125*
			*Glomus-MO-G23-VTX00222*
			*Glomus-sp.-VTX00165*
			*Glomus-intraradices-VTX00105*
			*Glomus-sp.-VTX00301*
Diversisporales	Diversisporaceae	*Diversispora*	*unclassified_g__Diversispora*
Paraglomerales	Paraglomeraceae	*Paraglomus*	*unclassified_g__Paraglomus*
			*Paraglomus-Glom-1B.13-VTX00308*
Archaeosporales	unclassified_o_ Archaeosporales	*unclassified_o_* *Archaeosporales*	*unclassified_o__Archaeosporales*

**Table 5 jof-10-00286-t005:** Morphological taxonomic identification and ecological parameters of AMF.

Phylum Glomeromycota Class Glomeromycetes Orders (4)	Families (6)	Genera (8)	Regions				IF	RA	IV	Dominance Rank
			ZN	GY	JH	DL				
Glomerales	Glomeraceae	*Glomus*	+	+	+	+	33.88%	91.36%	62.62%	Dominant genus
		*Rhizophagus*	+				1.22%	3.16%	2.19%	Rare genus
		*Septoglomus*	+				0.82%	2.11%	1.46%	Rare genus
Diversisporales	Gigasporaceae	*Scutellospora*	+		+	+	6.12%	19.97%	13.05%	Common genus
	Acaulosporaceae	*Acaulospora*	+	+		+	10.41%	31.32%	20.86%	Common genus
	Diversisporales	*Diversispora*		+			2.04%	10.00%	6.02%	Rare genus
Paraglomerales	Paraglomeraceae	*Paraglomus*			+		1.63%	4.55%	3.09%	Rare genus
Archaeosporales	Ambisporaceae	*Ambispora*			+		6.94%	19.32%	13.13%	Common genus

Note: + represents the occurrence of this species in the area. AMF, arbuscular mycorrhizal fungi; DL, Dulan; GY, Guyuan; IF, isolation frequency, IV, importance value; JH, Jinghe; RA, relative abundance; ZN, Zhongning.

**Table 6 jof-10-00286-t006:** Identification of AMF molecular classification.

Phylum Glomeromycota Class Glomeromycetes Orders (4)	Families (8)	Genera (8)	Frequency of Occurrence (%)			
			ZN	GY	JH	DL
Glomerales	Glomeraceae	*Glomus*	6.52%	–	2.17%	–
	Claroideoglomeraceae	*Claroideoglomus*	2.17%	2.17%	–	–
Diversisporales	Gigasporaceae	*Scutellospora*	2.17%	–	2.17%	–
	Acaulosporaceae	*Acaulospora*	–	–	2.17%	–
	Diversisporaceae	*Diversispora*	–	2.17%	–	–
Paraglomerales	Paraglomeraceae	*Paraglomus*	15.22%	15.22%	21.74%	19.57%
Archaeosporales	Ambisporaceae	*Ambispora*	–	–	–	2.17%
	Archaeosporaceae	*Archaeospora*	–	–	4.35%	–

Note: –, the species does not occur in the region; AMF, arbuscular mycorrhizal fungi; DL, Dulan; GY, Guyuan; JH, Jinghe; ZN, Zhongning.

## Data Availability

The raw reads have been stored in the NCBI Sequence Read Archive (SRA) database (PRJNA863906), and submitted ribosomal RNA (rRNA) in GenBank (ON822093-ON822117).

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
