# Peer review of "Diversity of Arbuscular Mycorrhizal Fungi of the Rhizosphere of Lycium barbarum L. from Four Main Producing Areas in Northwest China and Their Effect on Plant Growth"

_jof, 2024, doi:10.3390/jof10040286_

Round 1
Reviewer 1 Report
1. The work is devoted to the current problem of increasing plant productivity using AMF, which would significantly reduce the use of inorganic fertilizers and improve the quality of agricultural products. 2. The results of the work are original and distinguished by high scientific novelty, obtained using modern molecular genetic methods for identifying mycorrhizal fungi, as well as classical methods for assessing the intensity of mycorrhizal infection. It should be especially noted that the authors carried out a cycle of work, unique in complexity and labor intensity, to obtain spores of mycorrhizal fungi and artificially inoculate the roots of L. barbarum with them. 3. The manuscript of the article contains a sufficient amount of illustrative materials and tables that give a complete picture of the data obtained. 4. The authors discuss their findings in detail. 5. I believe that the materials of this work, when published, will be of interest to a wide range of specialists.
As comments, I would like to note the following. 1. The article, which ends with a discussion, looks, in my opinion, unfinished; it clearly lacks at least a brief conclusion. 2. One of the important, if not the main, result of the work is the demonstration by its authors of the positive effect of AMF on the development of the root system, the above-ground part of L. barbarum. However, the work does not contain data on how much percent the weight and size of plants increase when inoculated with AMF relative to the control. It is this indicator that shows the effectiveness of using AMF to increase productivity!
I consider it possible to publish the manuscript, preferably taking into account the comments made.
As comments, I would like to note the following. 1. The article, which ends with a discussion, looks, in my opinion, unfinished; it clearly lacks at least a brief conclusion. 2. One of the important, if not the main, result of the work is the demonstration by its authors of the positive effect of AMF on the development of the root system, the above-ground part of L. barbarum. However, the work does not contain data on how much percent the weight and size of plants increase when inoculated with AMF relative to the control. It is this indicator that shows the effectiveness of using AMF to increase productivity!
Author Response
1. The article, which ends with a discussion, looks, in my opinion, unfinished; it clearly lacks at least a brief conclusion.
Answer: I will add a conclusion at the end.
2. One of the important, if not the main, result of the work is the demonstration by its authors of the positive effect of AMF on the development of the root system, the above-ground part of L. barbarum. However, the work does not contain data on how much percent the weight and size of plants increase when inoculated with AMF relative to the control.
Answer: I added the supporting material in Table S5-7 and increased the values in the words (lines 407-413).
Reviewer 2 Report
This study characterizes the AMF communities associated with a plant that is distributed in four regions of China that contrast in environmental and edaphic conditions. It also uses two methodological approaches, morphology and DNA of the AMF spores, for its study, which is novel and provides robust exploration for the study of these symbionts. Perhaps something that was missing was giving more weight to the properties of the soils and the characteristics of the environments in the four areas studied to better explain the differences or similarities between the AMF communities, but this could well be interesting work in the future.
All my comments, observations and suggestions are expressed in the text of the manuscript. These were noted on the form I previously filled out for this review. Authors will be able to view each in the attached pdf text file. Please see in detail in the text, line by line, each and every one of my comments and observations.

Author Response
We are honored to receive your recognition for this study. I will answer the questions in the article one by one, and sort out the following.
Q1、Line 4 delete “Parameters”.
Line 18 delete “statistics”.
Line 19 delete “statistically”.
Line 351 delete “The AMF spores were isolated by wet sieve decantation and sucrose centrifugation”.
Line 429 / line 431 delete “.’.
Q2、Line 15 “physicochemical” change to “physical and chemical”.
Line 22 “saplings” change to “seedlings”.
Line 34 / line 41 “It” change to “it”.
Line 58 “materials” change to “nutrients”.
Line 114 Table “Soil properties” change to “Soil characteristics”.
Line 114 Table “Lycium Barbarum” change to “Lycium barbarum”.
Line 290 “infestation“ change to “colonization”.
Line 310 “Identification of the roots AMF OTU species in four regions of “Lycium barbarum” change to “Identification of AMF OTU species in Lycium barbarum roots”.
Q3、Line 18 “To analyze the morphology“ by Illumina? please improve writing.
Answer: By using Melzer's reagent to analyze the morphology. So I rewrite “By using Melzer's reagent to analyze the morphology and using Illumina Miseq sequencing technology to analyze molecular identification of AMF communities”.
Q4、Line 19 “among the four typical L. barbarum planting areas” which four areas?, improve writing.
Answer: among the four typical L. barbarum planting areas (Zhongning, Guyuan, Jinghe, Dulan).
Q5、Line 19 “L. barbarum“ put in italics.
Line 475 “Glomus” put in italics.
Q6、Line 24 “were inoculated with Paraglomus VTX00375” It is not clear whether this is an inoculation experiment or a study of AMF communities in the field. You should improve the clarity of the abstract in general.
Answer: This is an inoculation experiment. “were inoculated with Paraglomus VTX00375 in the pot experiment”.
Q7、Line 29 “Lycium barbarum; arbuscular mycorrhizal fungi (AMF)” use different words that are not already included in the title.
Answer: I changed them to” growth promotion; fungi”.
Q8、Line 35 / line 40 / line 56 / line 57 / line 62/ line 72 / line 81 / line 96 / line 97 / line 99 / line 156 / line 458 / line 476 / line 521 check a space
Q9、Line 36 / line 40 “L. barbarum” change to “Lycium barbarum”.
Q10、Line 44-47 “Current studies on native L. barbarum primarily focus on the effects of environmental factors such as climatic conditions and soil and water quality, but there is a lack of sufficient knowledge. For example, AMF species of research on rhizosphere microorganisms and molecular mechanisms of AMF colonization“ check the wording.
Answer: Current studies on native L. barbarum primarily focus on the effects of environmental factors such as climatic conditions and soil and water quality, but there is a lack of research on the Arbuscular mycorrhizal fungi (AMF) species of rhizosphere microorganisms and the molecular mechanism of AMF colonization.
Q11、Line 54 “Arbuscular mycorrhizal fungi (AMF)” even this sentence defines an acronym that was used before, please correct this.
Answer: I put its explanation to line 46.
Q12、Line 67 samples, but it is difficult and has limitations with non-sporulating AMF.
Answer: For the sample problem, we usually separate and select the AMF of the sample. We will discard the samples without AMF spores.
Q13、Line 75 Evaluations based on morphological traits with molecular.
Answer: Classic taxonomy is usually another way of saying that morphological identification.
Q14、Line 88 / line 511 Author information supplement complete.
Q15、Line 101 The influence of AMF on these plants or vice versa? I suggest placing the objectives of this research directly and explicitly in this section, in addition to adding a working hypothesis.
Answer: Here I briefly increase the experimental objectives and significance.“As a Taoist medicinal herb that is the result of specific geographic and ecological conditions, it is worth exploring the amount of influence of AMF on the growth of Lycium barbarum. The growth-promoting effects of different AMF may be different. Therefore, it is of great significance to explore the dominant species that can promote the growth of Lycium barbarum”.
Q16、Line 106 Between 20 and 40? They were the first 20cm discarded.
Answer: For the sampling method, our sampling is relatively deep, mainly to obtain the soil samples around the lateral roots of the plant.
Q17、Line 114 Complete with information that describes details of the table.
Answer: I changed it to the geographical location, environmental conditions and soil types of sample collection.
Q18、Line 126 “stained using trypan blue”. Please cite the reference of this methodology.
Answer: I have added references.
Q19、Line 136 What was the trypan blue stain for?
Answer: Alkali dissociation occurs because KOH treatment is required in the first step of this experimental procedure, and trichothecene blue which is also known as trypan blue staining.
Q20、Line 145 This figure goes in the results section.
Answer: I put it into 3.1.
Q21、Line 160-161 I have revised it to the original reference.
Q22、Line 162 Please indicate where the reference spores are deposited, were permanent slides made, where are these deposited?
Answer: As mentioned in the article, the spores we refer to are based on the descriptions and pictures provided by professional websites and books, and our own permanent slides are kept in the laboratory.
Q23、Line 163 Please indicate where the obtained sequences are deposited?
Answer: In the final data statement, we mentioned that the obtained sequences were uploaded to the NCBI database in GenBank(ON822093-ON822117).
Q24、Line 260 / line 448-450 Briefly mention why you decided to use OTU and not ASV.
Answer: The clustered sequence is then designated as an Operational Taxonomic Unit (OTU), with a typical identity threshold of 97% or higher during clustering. The current completeness of the AMF database is limited, making OTU a significant choice. Classifying sequenced AMF species without relying on the AMF database is more conducive to discovering new species or those with low abundance in sequencing. However, Amplicon Sequence Variant (ASV) analysis eliminates low-quality sequences and sets the identity threshold at 100% during comparison, which may result in misclassification and elimination of rare species in sequencing.
Q25、Line 265 Briefly mention why you decided to use this multivariate method, what advantage does it have over others like the NMDS?
Answer: Sequencing data are primarily categorized as OTU. PCA is mainly utilized for analyzing the sample OTU of microbial community research, and the original OTU abundance data is remapped through dimensionality reduction. PCoA and NMDS primarily focus on studying the Beta diversity analysis of microbial communities, which is based on reducing the dimensionality of similarity distances among various samples. The samples in this study mainly reflect alpha diversity, specifically species abundance and differences. Beta-diversity analysis, which measures changes in species on a spatio-temporal scale, was not conducted.
Q26、Line 278 And how were the ecological parameters of the community structure analyzed?
Answer: Through sequencing data, we can understand the composition, structure and evolution of microbial communities. The classification of microorganisms was determined by BLAST comparison results. After the comparison and annotation, the list and relative abundance information of microbial species in different samples can be obtained. In order to display the composition and structure of microbial communities more intuitively, biodiversity analysis tools such as. This paper mainly analyzes the root and stem-related indexes of Lycium barbarum.
Q27 Line 352 I add “morphological”.And the pictures in the supporting materials (Figures S1 – S3).
Q28 Line 373 Table6 Why not do this analysis and the one in the previous table at the species level? I suggest adding the species with their taxonomic descriptors?
Answer: The AMF classification system lacks a unified scale extent. The special MaarjAM AMF fungi database (http://www.maarjam.botany.ut.ee/) serves as the virtual, molecular-level classification for AMF. However, its implementation is not accurate.
Q29 Line 408-412 All this corresponds to the discussion.
Answer: I am sorry may be my understanding of the problem, I think he is consistent with the text of the discussion of 502-520.
Q30 Line 425 I am properly relocating this reference.
Q31 Line 540 Add a final paragraph highlighting your main contributions as a final conclusion.
Answer: I add the conclusion.
Q32 line 114 Table1 Please use a classification such as international WRB or its possible equivalence.
Answer: The soil type has been modified according to WRB.
Q33 line 416 Please add in the figure or in the text, the statistical parameters of the effects of the treatments in the anova.
Answer: I added supporting materials Table S5-7, and increased the value in the word (lines 407-413).
Q34 line 467 Comparisons with what has been reported for AMF in other arid regions of the world are desirable to incorporate.
Answer: In the following, there are some comparisons with this kind of results. If you don 't think it is enough, I will add this part later.